# Elucidating the inhibitory mechanism of Zika virus NS2B-NS3 protease with dipeptide inhibitors: Insights from molecular docking and molecular dynamics simulations

**Shahid Ullah**[1]*, **Farhan Ullah**[1], **Wajeeha Rahman**[1], **Anees Ullah**[1], **Sultan Haider**[1], **Cao Yueguang**[2]*

**1** S-Khan Lab, Mardan, Khyber Pakhtunkhwa, Pakistan, **2** Huazhong University of Science and Technology Union Shenzhen Hospital, Nanshan, Shenzhen, China

* drskbioch@gmail.com (SU); caoyueguang1969@hotmail.com (CY)

**Data Availability Statement:** All relevant data are within the paper.

## Abstract

Microcephaly, Guillain-Barré syndrome, and potential sexual transmission stand as prominent complications associated with Zika virus (ZIKV) infection. The absence of FDA-approved drugs or vaccines presents a substantial obstacle in combatting the virus. Furthermore, the inclusion of pregnancy in the pharmacological screening process complicates and extends the endeavor to ensure molecular safety and minimal toxicity. Given its pivotal role in viral assembly and maturation, the NS2B-NS3 viral protease emerges as a promising therapeutic target against ZIKV. In this context, a dipeptide inhibitor was specifically chosen as a control against 200 compounds for docking analysis. Subsequent molecular dynamics simulations extending over 200 ns were conducted to ascertain the stability of the docked complex and confirm the binding of the inhibitor at the protein's active site. The simulation outcomes exhibited conformity to acceptable thresholds, encompassing parameters such as root mean square deviation (RMSD), root mean square fluctuation (RMSF), ligand-protein interaction analysis, ligand characterization, and surface area analysis. Notably, analysis of ligand angles bolstered the identification of prospective ligands capable of inhibiting viral protein activity and impeding virus dissemination. In this study, the integration of molecular docking and dynamics simulations has pinpointed the dipeptide inhibitor as a potential candidate ligand against ZIKV protease, thereby offering promise for therapeutic intervention against the virus.

## Introduction

Zika virus (ZIKV) is a flavivirus that belongs to the Flaviviridae family [1,2]. The virus was transmitted by mosquitoes and led to a pandemic and a public health disaster in 2016 [3–5]. The most prevalent mode of transmission of ZIKV is through the bites of infected mosquitoes, *Aedes aegypti*. It was first isolated from a non-human monkey in 1947 and then from

**Funding:** The author(s) received no specific funding for this work.

**Competing interests:** The authors have declared that no competing interests exist.

mosquitoes in Africa a year later [6]. Human ZIKV infections remained low for more than a century before spreading to the Pacific and the Americas[7–9]. The arrival of ZIKV in Brazil in 2015 marked the rapid spread of the virus in the Americas, where it had previously been limited to isolated cases in Africa and Asia [10,11]. ZIKV preferentially infects neural stem cells, astrocytes, oligodendrocyte precursor cells and microglia, while neurons are less susceptible [12]. The genome of ZIKV undergoes a series of reorganization events, such as the loss and gain of the N-linked glycosylation site of the E protein, making it possible for them to spread and cause infection in different host types [2,5,13], and the mutagenic ability of the virus is potentially problematic [14,15]. ZIKV has an enveloped icosahedral virion with a 40 to 50 nm diameter. It comprises a non-segmented, single-stranded, positive-sense RNA genome and ten proteins, three of which are structural and seven nonstructural [16,17]. The most abundant protein on the virion is the E protein, which is involved in many steps of the viral cycle, including membranes, viral entry processes, assembly, fusion, and binding. Structure-based antiviral discovery and antigen identification for vaccine development primarily focus on these proteins [18]. According to the literature, ZIKV shares sequence similarity with Dengue virus and several other human flaviviruses [19,20]. ZIKV shared some common phylogenetic relationships from the nineteenth to twentieth centuries, as described by Oumar Faye et al [21].

The NS2B-NS3 viral protease is an attractive drug target against ZIKV due to its importance in viral replication and maturation. The ZIKV protease is a two-component protease formed by the protease domain of the N-terminal region of NS3 and the cytoplasmic region of the NS2B cofactor [22]. Structural studies of the west Nile virus (WNV) and dengue virus (DENV) proteases and new structural studies of the ZIKV protease have shown that the development of low molecular weight drugs targeting the active region is complex due to the negative charge of the active site of the protease [23–25]. Peptidomimetic inhibitors have been successfully developed, and some inhibitors generated from P1 to P3 residues of the substrate are active against WNV and DENV proteases [26–28]. According to the research of Li et al., in complex, the protease crystal structure with a dipeptide inhibitor, Acyl-KR-aldehyde (compound 1) is reported which can be used as a reference in future research [15]. The inhibitor establishes polar interactions with Asp83 of NS2B and Asp129 of NS3. The aldehyde group creates a covalent bond with NS3 of Serine 135. Hence, this dipeptide inhibitor, can serve as a reference drug for the development of protease inhibitors.

This study is based on the bioinformatics analysis of Zika virus protease in complex with a dipeptide inhibitor, aiming to identify more effective therapeutic targets than those offered by currently available drugs. Here, we introduce a methodology that combines pharmacophore modeling and structure-based virtual screening techniques to discover potential inhibitors for Zika virus control. The virtual screening process involved screening ligand-based compound libraries using a 3D similarity search approach. These libraries were specifically screened against FDA-approved and recommended medications for the treatment of Zika virus. Additionally, a comparative analysis was conducted against strong natural compounds that have already been predicted.

The aim of this study is to identify potential FDA-approved and natural inhibitors against the Zika virus that are currently available on the market. The investigation focused on predicting compounds that exhibit stronger binding affinities compared to already reported compounds. For this purpose, two cycles of similar methodologies were performed separately for FDA-approved and natural compounds to conduct a comparative analysis of their binding properties.

## Materials and methods

### Preparation of ligands

A ligand dipeptide inhibitor was used as a control for against FDA and natural compounds. The ligand structure was retrieved from PubChem (CID: 137348621). The ligand preparation was done by minimizing it using chemdraw 3D.

In comparisosn to the controlled ligand, the FDA library of 200 compounds was built using the ZINC database. Further for selecting top ten best compound Ligand-based virtual screening (LBVS) was performed using ligandscout.4. In order to conduct molecular docking experiments, the top 80 best hits of LBVS was ran in pyrx for molecular docking [29].

### Preparation of target proteins

The X-ray crystal structures of pathogenic proteins were retrieved from the RCSB Protein Data Bank The target 3D Zika virus protease structure, with ID 5H6V, was acquired from the Protein Data Bank (PDB). The retrieved protein contained 687 amino acids, with a resolution of 2.42 Å predicted through X-ray diffraction. The structure was obtained in PDB format and refined using Chimera. The chain B, ligands, ions, and water molecules were removed from the target protein. Subsequently, the refined structure as a macromolecule was prepared in PDBQT format using PyRx/AutoDock Vina for docking studies.

### Molecular dynamics simulation (MDS) study

The Molecular Dynamics (MD) simulations were performed using the GROMACS software, which utilized the AMBER force field [30,31]. The simulations were run for a total of 100 nanoseconds. For performing protein-ligand docking, through molecular dynamic simulation to evaluate the rigidity of binding characteristics of selected drugs with the target protein [32]. The work employed molecular dynamic (MD) simulations to forecast the binding efficacy of ligands in a physiological milieu [33]. The protein and ligand that chosen were subjected to minimization to eliminate unfavorable interactions, distorted molecular structures, and steric hindrances. In order to carry out an inspection, the paths were recorded at intervals of 100 picoseconds (ps) using a total of 2000 frames created for this run. The stability of the protein-ligand complex was evaluated by analyzing the Root mean square fluctuation (RMSF), Radius of gyration (Rg), and Root Mean Square Deviation (RMSD) over a specific time period [34].

## Results and discussion

### Molecular docking study

Molecular docking was carried against the 200 FDA compounds as well as for natural compound in comparsion to reference drug, a nature compound. After screening of 200 fda compounds available in the market, the best 6 compounds were selected. Similarly, in the already available nature compounds against zika virus the top 5 were analysed further in comparison to eachother, along with the reference drug (CID: 137348621).. The docking results using pyrx are mentioned in the **Table 1.**

In FDA best compounds, the highest binding affinity was reported for ZINC000000004351–7.5 kcal/mol while for ZINC000000005823 is -7 kcal/mol, ZINC000000002191 is -6.9 kcal/mol and for ZINC000000003876, ZINC000000003642 and ZINC000000004448 is -6.7 kcal/mol respectively. The best and stable results were reported for ZINC000000002191 and it was sent for studing post docking results through simulation.

**Table 1. The docking results from pyrx showing biniding affinity and root mean square deviation.**

| No. | Compound ID | Binding Affinity | RMSD |
|---|---|---|---|
| 1 | ZINC000000004351 | -7.5 | 1.748 |
| 2 | ZINC000000005823 | -7.0 | 1.491 |
| 3 | ZINC000000002191 | -6.9 | 1.328 |
| 4 | ZINC000000003876 | -6.7 | 1.695 |
| 5 | ZINC000000003642 | -6.7 | 1.695 |
| 6 | ZINC000000004448 | -6.6 | 1.034 |
| **Natural Products** | | | |
| 7 | (CID_5288209) | -7.9 | 1.032 |
| 8 | (CID_65094f) | -7.3 | 2.812 |
| 9 | (CID_72378) | -7 | 1.781 |
| 10 | (CID_253602) | -5.6 | 1.041 |
| 11 | (CID_2719) | -5.4 | 1.631 |
| 12 | Ref.drug(CID_137348621) | -5.0 | 1.713 |

Molecular docking was carried out between targeted viral protein of Zika virus (ZIKV protease) having PDB ID 56HV and ligand dipeptide ligand. The reference ligand was obtained from Pubchem with the CID-137348621. The ligand and protein was visualized using chemdraw and chimera in order to find best and reliable docking score. The Fig 1. presents the targeted protein and 2D/3D structure of ligand.

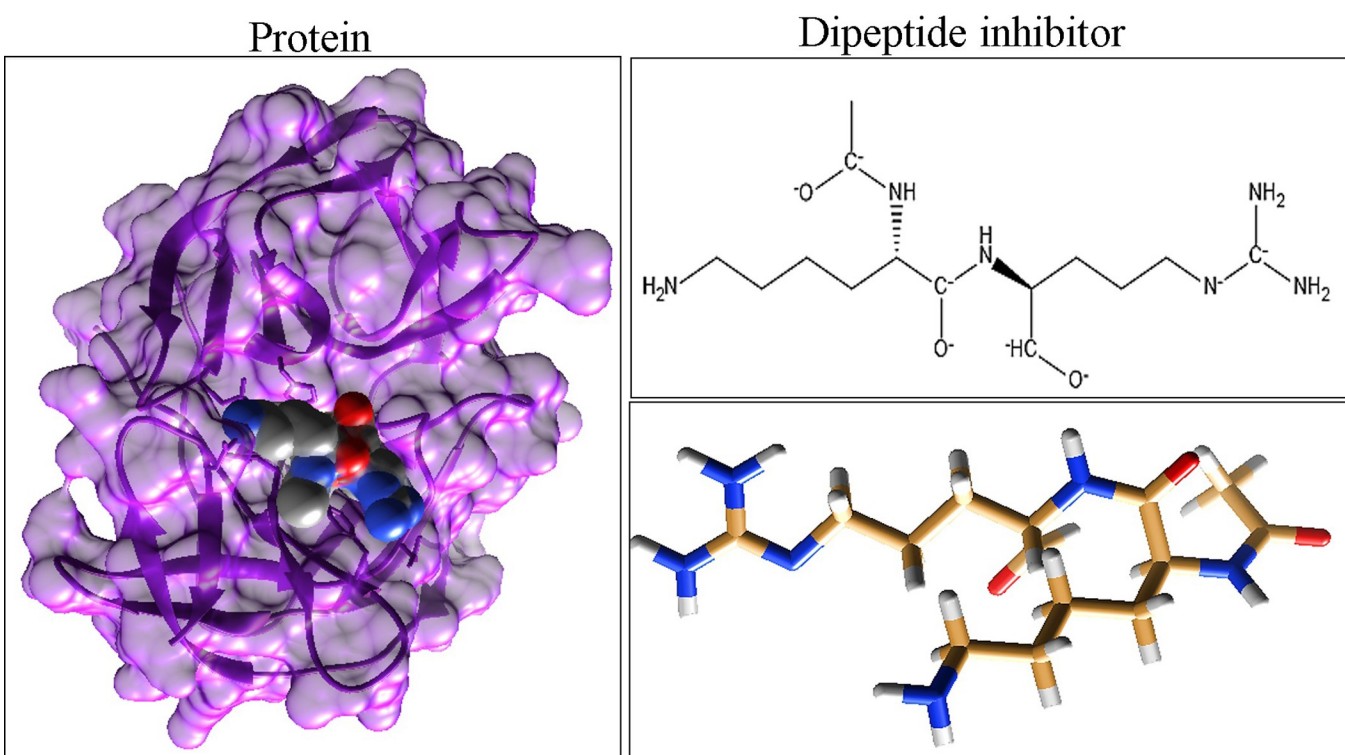

**Fig 1. Presenting the targeted viral protein (5H6V) of Zika virus, The 2D structure of ligand with its Pubchem ID and 3D structure of targeted ligand (Dipeptide inhibitor).**

The binding interaction of dipeptide inhibitor used as a reference drug to treart zika virus was studies using Autodock vina and pyrx. The binding energy of this reference drug was -5.0 kcal/mol. Following the docking, the ligand-receptor receptor complex was carried out for interaction analysis. The study of H-bonds and other interacting residues was perfomed by using BIOVIA discovery studio. For all bonds, the angle was set between 150 and 180 degrees. The observed binding residues are presented in Fig 2.

Based on binding affinity and interacting residues in contrast of ref.drug, the docked complex of ZINC000000002191-5H6V with the energy of -6.9kcal.mol was selected as the compound for molecular dynamic simulation. Fig 3 is presenting the 3D and 2D interaction of the best selected docked complex.

## MD simulation

Molecular dynamics (MD) simulation modeling is a commonly employed technique for studying the kinetics and complexes stability of ligand-receptor under physiological conditions [35]. This technology is applied to investigate the stabilization of protein-ligand complexes, anticipate binding modes, and explore potential interactions during the process of ZIKV protease binding to inhibitors that are especially designed for treating Zika virus. A molecular dynamic simulation was conducted on the ZIKV protease protein and the docked complex of the best selected ligand ZINC000000002191 (Tolmetin) for a time of 100 ns to perform post-dock

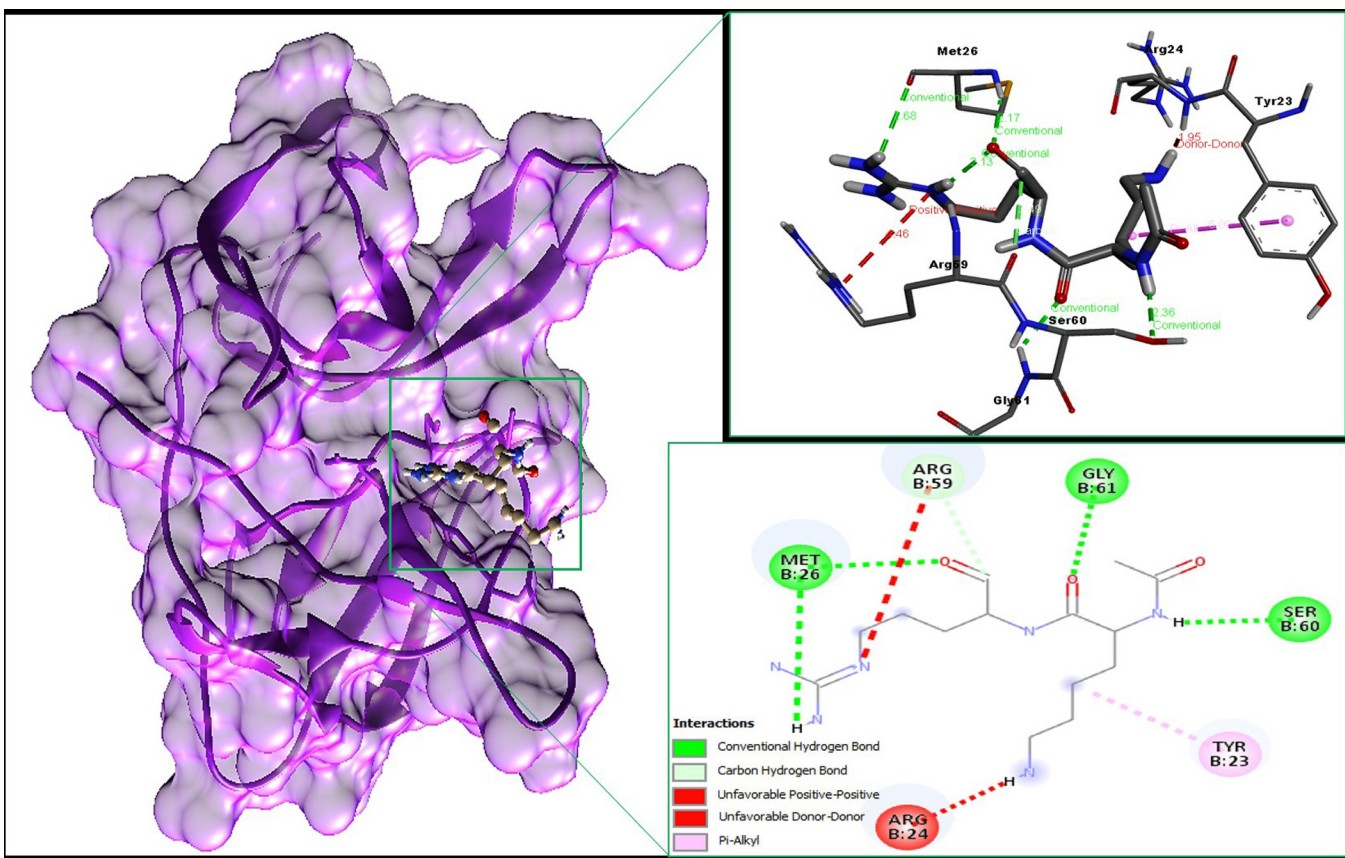

**Fig 2. Illustration of the binding residues of docked complex of targeted protein of Zika virus and dipeptide inhibitor.**

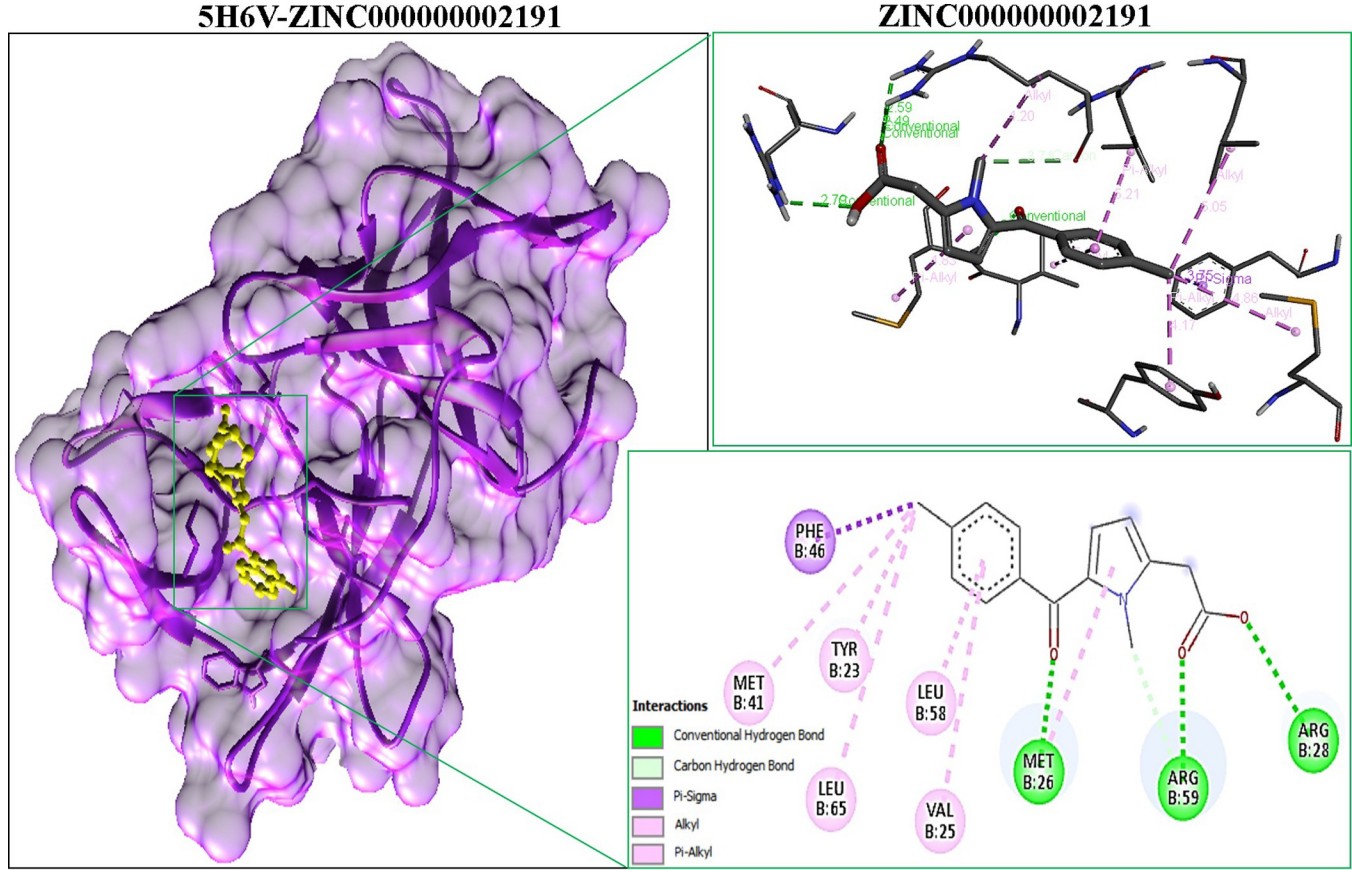

**Fig 3.** Binding residues of Zika virus protein and dipeptide inhibitor: (A) 3D protein-ligand complex, (B) Ligand and interacting residues with bond distances and types.

verification. The changes in interaction of docked protein-ligand complex at different interval in shown in Fig 4.

## MD simulation

**Root Mean Square Deviation (RMSD).** The RMSD (root mean square deviation) was employed in dynamic modeling to investigate the structural stability of complex of ZIKV protease (5H6V) and the complex formed by ligand ZINC000000002191 and protein. The RMSD of ligand is determined by aligning the structure obtained from MD simulation with the initial frame trajectory [36]. **Fig 5.** indicates the RMSD of Tolmetin (ZINC000000002191) complex with highest velocity was observed at 32.94ns of 2.65Å. The RMSD throughout the simulation lied between 0.5 Å and 2.7 Å.

**Root Mean Square Fluctuation (RMSF).** The initial Root Mean Square Fluctuation (RMSF) observed for GLU 17 was 2.62 Å, with a maximum RMSF value of 3.88 Å. The residues VAL 155, PHE 46, GLU 103, GLY 104, LYS 142, and CYS 143 demonstrate an RMSF greater than 3 Å. Therefore, the chosen ligand had a moderate root mean square fluctuation (RMSF) value, indicating its stability against the targeted protein. **Fig 6.** illustrates the generated distribution across the protein structure. Higher root mean square fluctuation (RMSF) results indicate that these particular residues undergo greater degree of fluctuations or mobility during the data gathering procedure. This enhanced flexibility indicates particular regions

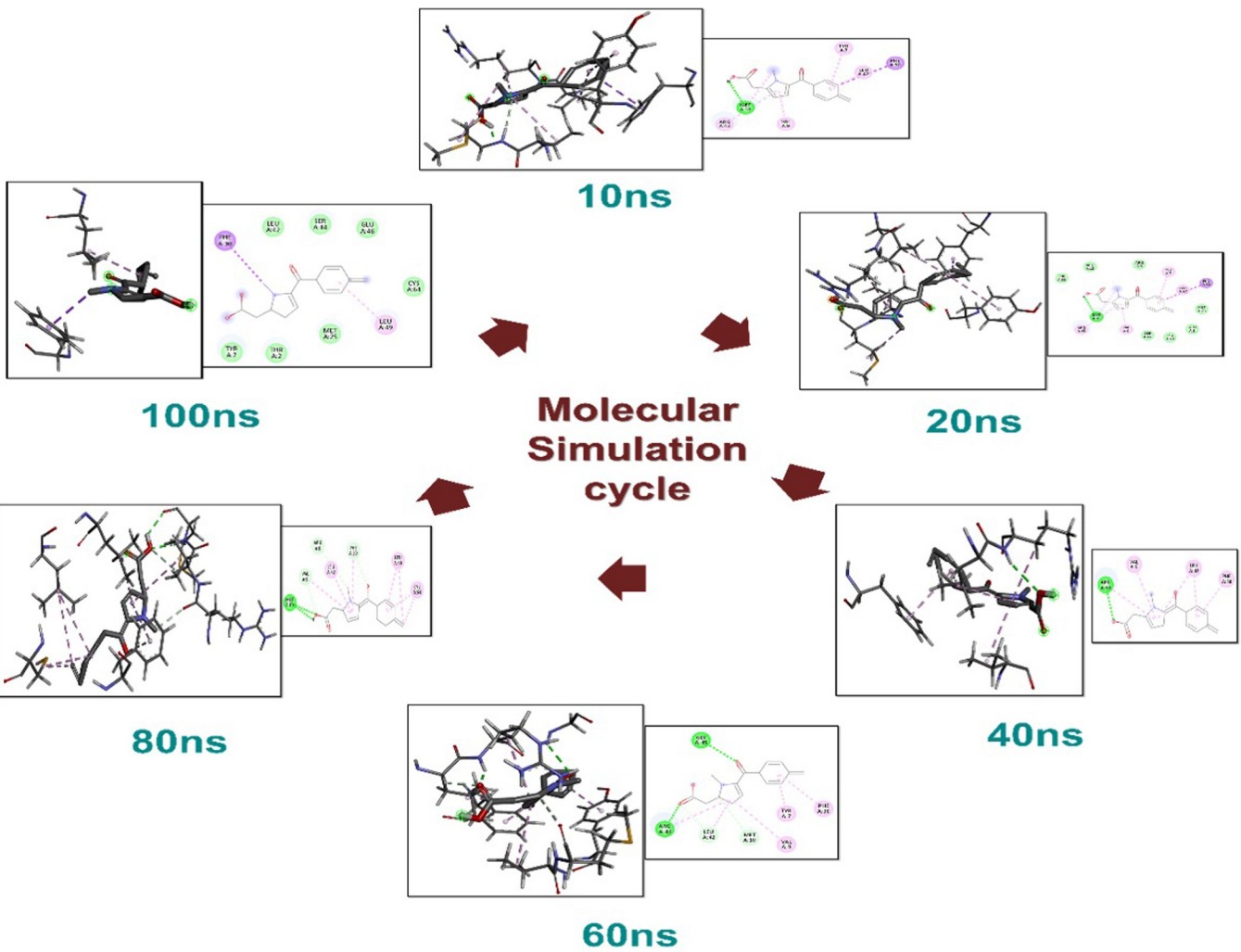

**Fig 4. The changes for simulation have been shown at different nanoseconds.**

of the protein that exhibit a high degree of dynamism or are prone to undergo conformational alterations.

### Radius of gyration (Rg) and Principal component analysis (PCA)

Investigating the Radius of Gyration (Rg), an important indicator for assessing the compactness and mobility of biomolecules. By subjecting the protein of interest to intensive computer simulations, we found a maximum radius of gyration of 15.36Å at a time of 53.62ns. **Fig 7** is presenting Radius of gyration of best docked complex of ZINC000000002191-5H6V. While the Rg and PCA is shown in **Fig 8A and 8B.**

Lastly, the binding-free energy of a ligand to a protein is estimated by employing MMPBSA and MMGBSA.These calculation are presented in the Table 2.

### Conclusion

This study identified a dipeptide inhibitor as a potential candidate for targeting the NS2B-NS3 viral protease to combat Zika virus (ZIKV) infection. Molecular docking and dynamics

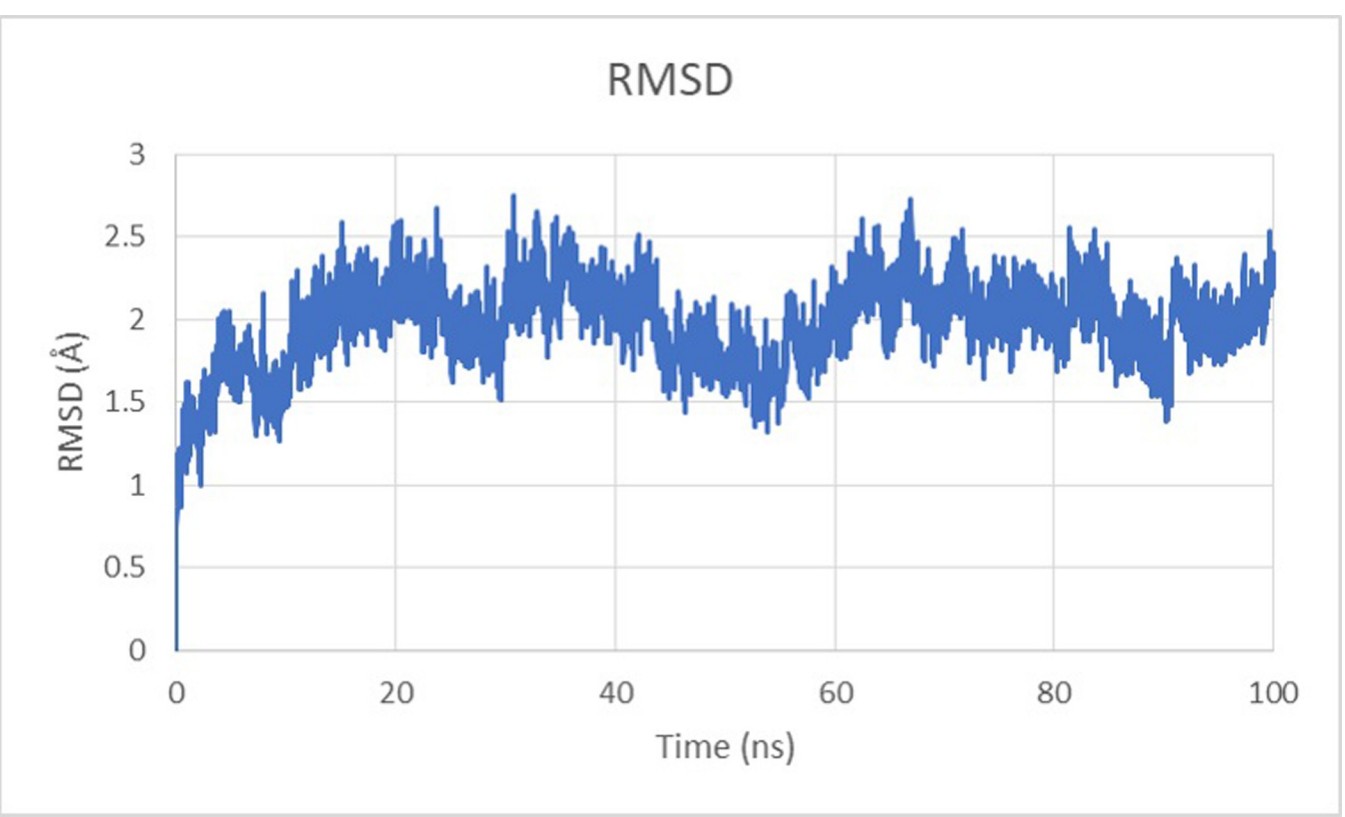

**Fig 5. The graphical representation of ZIKV protease and Tolmetin complex RMSD at 100ns.** The maximum RMSD (Å) velocity of 2.65 has been observed at 32.94ns.

simulations revealed stable binding sites, further analyzed by MD simulations using normal mode analysis. Physical parameters such as RMSD, RMSF, radius of gyration (Rg), and principal component analysis (PCA) characterized the stability of the protein-ligand complex,

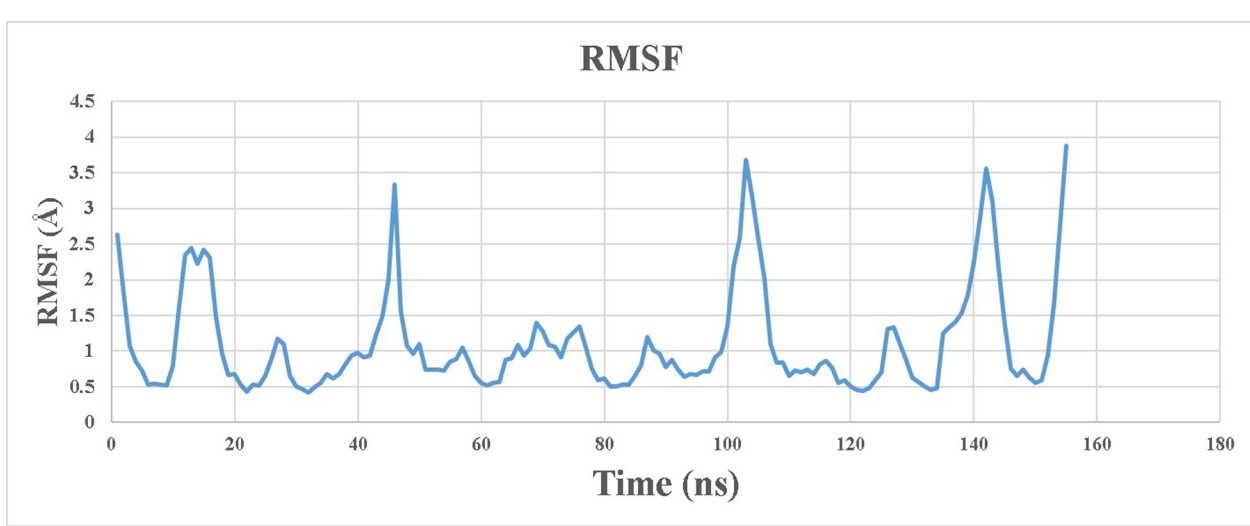

**Fig 6. RMSF of Tolmetin complex at 100ns.** The ideal RMSD (Å) has been observed of maximum velocity at 3.88Å.

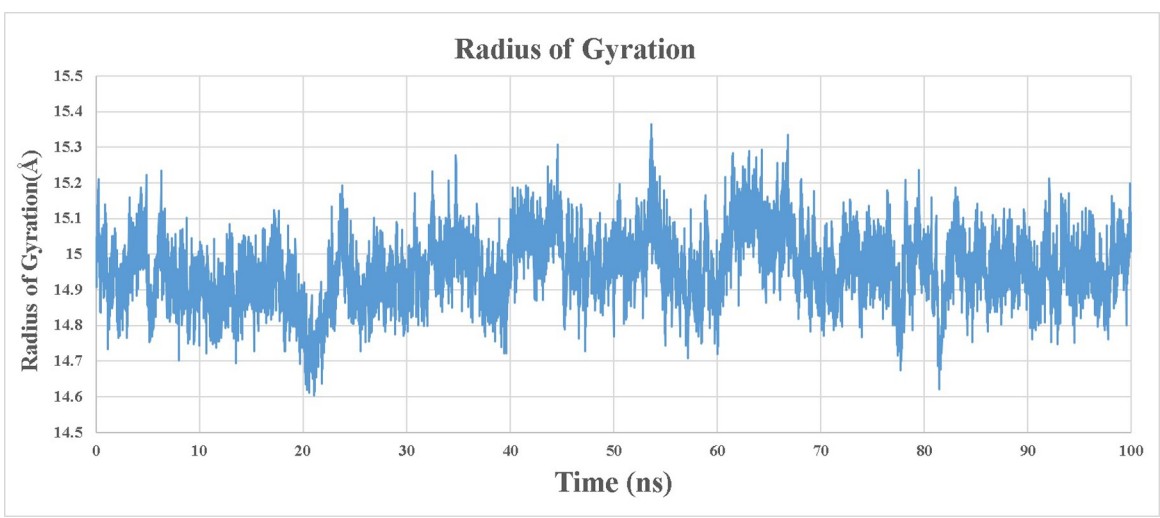

**Fig 7. Radius of gyration (Rg) of ZINC000000002191-5H6V.**

showing consistent results. Additionally, binding energy values were determined using MMPBSA and MMGBSA during the simulations..The graphs highlighted ligand-receptor contacts, including biomolecules, and identified individual amino acids involved in these interactions. The complementary in-silico results suggest further in vivo and in vitro investigations are needed to refine the anti-ZIKV properties of this inhibitor. Broadening this study to other stages of the virus's life cycle or other bacteria would be beneficial. Considering the complexities of ZIKV infection, such as microcephaly, Guillain-Barré syndrome, and sexual transmission, future studies should assess the dipeptide inhibitor's effectiveness in reducing these complications or targeting other viral proteins. Despite the urgency for new ZIKV treatments due to the lack of FDA-approved medications or vaccines, incorporating pregnancy into pharmacological screenings adds complexity and extends safety and efficacy evaluations. Future research should address these challenges. To overcome current limitations, computational methods like machine learning could enhance the precision and speed of virtual screening and molecular dynamics simulations. Confirming computational results with in vitro and in vivo

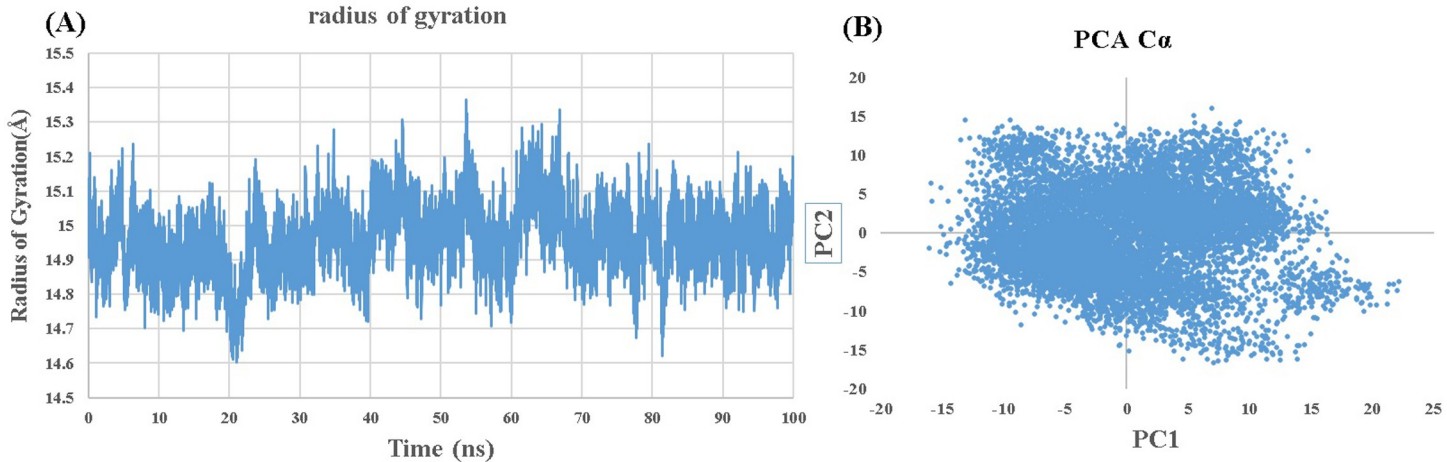

**Fig 8.** For Tolmetin (ZINC000000002191) as a potent ZIKV protease (5H6V) inhibitor: (A) Radius of gyration and (B) Principal Component Analysis (PCA).

**Table 2. Presenting the free binding energies using MMGBSA and MMPBSA.**

| Energy Component | Average | Std. Dev. | Std. Err. of Mean |
|---|---|---|---|
| VDWAALS | -38.6544 | 3.8307 | 1.2114 |
| EEL | -6.0282 | 3.6011 | 1.1388 |
| EPB | 19.6489 | 3.2414 | 1.0250 |
| ENPOLAR | -21.9922 | 1.3044 | 0.4125 |
| EDISPER | 39.2225 | 2.0259 | 0.6406 |
| DELTA G gas | -44.6825 | 5.4451 | 1.7219 |
| DELTA G solv | 36.8792 | 4.1039 | 1.2978 |
| DELTA TOTAL | -7.8033 | 2.9894 | 0.9453 |

studies would provide crucial insights into the dipeptide inhibitor's efficacy against ZIKV. Shortly, this study marks significant progress in identifying potential ZIKV treatments. Future research should expand the scope, address limitations, and validate findings experimentally to apply these computational insights in clinical settings.

## Acknowledgments

We are thankful to the S. Khan Lab (https://www.habdsk.org/) [37] for providing the facilities to conduct this research. We appreciate the work done at the S. Khan Lab on the Molecular Binding Energy Database (https://www.pbed.habdsk.org/), which is a significant contribution to the scientific community in this research area.

## Author Contributions

**Software:** Farhan Ullah.

**Supervision:** Shahid Ullah, Cao Yueguang.

**Validation:** Farhan Ullah, Wajeeha Rahman, Anees Ullah, Sultan Haider, Cao Yueguang.

**Visualization:** Shahid Ullah, Farhan Ullah, Cao Yueguang.

**Writing – original draft:** Shahid Ullah.

**Writing – review & editing:** Shahid Ullah.

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
