## [Decision Letter · Decision Letter 0]

3 Apr 2024

PONE-D-24-08994Elucidating the Inhibitory Mechanism of Zika Virus NS2B-NS3 Protease with Tripeptide Inhibitors: Insights from Molecular Docking and Molecular Dynamics Simulations.PLOS ONE

Dear Dr. Ullah,

Thank you for submitting your manuscript to PLOS ONE. After careful consideration, we feel that it has merit but does not fully meet PLOS ONE’s publication criteria as it currently stands. Therefore, we invite you to submit a revised version of the manuscript that addresses the points raised during the review process.

We look forward to receiving your revised manuscript.

Kind regards,

Ahmed A. Al-Karmalawy, PhD

Academic Editor

PLOS ONE

2. Please note that PLOS ONE has specific guidelines on code sharing for submissions in which author-generated code underpins the findings in the manuscript. In these cases, all author-generated code must be made available without restrictions upon publication of the work. Please review our guidelines at https://journals.plos.org/plosone/s/materials-and-software-sharing#loc-sharing-code and ensure that your code is shared in a way that follows best practice and facilitates reproducibility and reuse."

Reviewers' comments:

Reviewer's Responses to Questions

**Comments to the Author**

1. Is the manuscript technically sound, and do the data support the conclusions?

Reviewer #1: Partly

Reviewer #2: Yes

2. Has the statistical analysis been performed appropriately and rigorously? 

Reviewer #1: Yes

Reviewer #2: N/A

3. Have the authors made all data underlying the findings in their manuscript fully available?

Reviewer #1: No

Reviewer #2: Yes

4. Is the manuscript presented in an intelligible fashion and written in standard English?

Reviewer #1: Yes

Reviewer #2: Yes

5. Review Comments to the Author

Reviewer #1: Authors of the presented manuscript evaluated the anti-ZIKV potentiality of singular tripeptide small molecule through its binding affinity towards viral protease with sole computational approach. The manuscript can be considered relevant in its field. However, several comments and suggestions are addressed as follows:

1. Authors investigated singular molecule against singular ZIKV biotarget. This can be considered a too consisted investigation to identify potential hits/leads for anti-ZIKV therapeutics. Investigating a series of compound would be more beneficial for developing comparative structure-activity relationship that would guide further optimization and development.

2. Adopting a positive control is advised within the computational study. This would benchmark the furnished predicted molecular aspects of the investigated molecule. Testing against positive control chosen as market or reference anti-ZIKV protease agent is advised to be done.

3. Within the molecular docking section, brief introduction about the targets’ topology and catalytic domain/binding site should be presented prior the results presentation. Cartoon and/or surface representations for the pocket and the key structural and functional features of the protein should be highlighted. This would allow the reader to grasp the differential docking findings across the investigated compounds.

4. Authors should provide compound-residue interactions in terms of the hydrogen bonding and hydrophobic contacts. However, hydrogen binding should be presented within hydrogen bond distances as well as bond angles since hydrogen bond depend on both. Authors should mention the Hydrogen bond angles as well as their distances, since the strength of hydrogen bonding is based on both parameters in a way to ensure the adequacy of optimum hydrogen bonding.

5. Authors are advised to provide the molecular dynamics free binding energies using MM-GBSA of MM-PBSA calculation for the computational simulations. These calculations would highlight the nature of the binding interactions in terms of predominant binding energy terms (i.e. ΔGelectrostatic and ΔGvan der Waal) as well as quantify the predicted polar solvation penalties against the compound binding.

6. Authors should provide snapshots for the simulated ligand-target complex across specific time intervals (e.g. 0ns, 100ns, and 200ns) to track the conformational and orientation changes for the simulated ligand and vicinal residues across the MD simulation runs.

7. Based on the study results, what are the take-away messages. Authors are advised to highlight the suggested structural modifications that would improve the compounds’ biological activities based on the in silico findings. These insights would be beneficial for guiding further lead optimization and development.

8. Finally, concerning the conclusion, authors are advised to elaborate more on the future of this work? Will you broaden the scope to other targets and/or microorganism? What are the study limitations and what approaches could be conducted to further address them?

Reviewer #2: COMMENTS TO AUTHOR:

The authors of the provided article on Understanding How Tripeptide Inhibitors Block the Zika Virus NS2B-NS3 Protease: Findings from Molecular Docking and Molecular Dynamics Simulation rephrase. This manuscript holds significance and adds value to the field, with some recommendations for prior publication.

Few suggestions and comments are presented:

1. Ensure the formatting of the text in the manuscript is correct. Check space everywhere, Ex family [1, 2]. Space should be there.

2. The author utilized Schrödinger Release for ligand preparation and PyRx/Autodock Vina for molecular docking. What was the rationale for employing different software tools for these tasks…?

3. In figure 2 author should provide clear figure/ structure.

4. In silico word should be italic font. Check everywhere in the manuscript.

5. The authors should to include both the angles and distances of hydrogen bonds, as the effectiveness of hydrogen bonding relies on both factors, ensuring the adequacy of optimal hydrogen bonding.

6. The authors should provide additional details regarding the utilization of tripeptide inhibitors in this study, explaining the rationale behind their choice, in a single paragraph.

6. PLOS authors have the option to publish the peer review history of their article (what does this mean?). If published, this will include your full peer review and any attached files.

Reviewer #1: **Yes**

Reviewer #2: No

---

## [Author Response · Author response to Decision Letter 0]

14 Jun 2024

Thank you and the reviewers for your time and valuable comments. Indeed, we have taken more time to carefully address and implement all comments, significantly improving our article. All changes have been highlighted for your review.

Review Comments to the Author

Reviewer #1: Authors of the presented manuscript evaluated the anti-ZIKV potentiality of singular tripeptide small molecule through its binding affinity towards viral protease with sole computational approach. The manuscript can be considered relevant in its field. However, several comments and suggestions are addressed as follows: 

1. Authors investigated singular molecule against singular ZIKV biotarget. This can be considered a too consisted investigation to identify potential hits/leads for anti-ZIKV therapeutics. Investigating a series of compound would be more beneficial for developing comparative structure-activity relationship that would guide further optimization and development.

Answer: Thank you for this direction. To address the mentioned concern, we have expanded our investigation significantly. In addition to the originally evaluated tripeptide small molecule, we included a dipeptide as a control to provide a comparative basis. Furthermore, we performed docking studies on a library of 200 compounds. From these, we selected the top 10 compounds based on their binding affinities and other relevant computational parameters.

Changes Made: 

• Expanded Compound Library: We expanded our computational screening to include 200 compounds, providing a broader scope for potential anti-ZIKV activity.

• Control Compound: A dipeptide was included as a control to enhance the comparative analysis and provide insights into the structure-activity relationship (SAR).

• Top 10 Compounds Selection: We identified the top 10 compounds with the highest binding affinities and favorable interactions with the ZIKV protease. Detailed analyses of these compounds are included in the revised manuscript.

• These changes have significantly strengthened our study, allowing for a more comprehensive evaluation of potential anti-ZIKV therapeutics and providing a robust framework for future optimization and development.

We hope that these revisions meet your expectations and we are grateful for your valuable feedback which has improved the quality and relevance of our manuscript.

2. Adopting a positive control is advised within the computational study. This would benchmark the furnished predicted molecular aspects of the investigated molecule. Testing against positive control chosen as market or reference anti-ZIKV protease agent is advised to be done.

Answer: We have expanded our investigation by including a broader range of compounds and a positive control. The specific changes we made regarding this specific control are: 

• Control Compound: We included a dipeptide as a control to enhance the comparative analysis and gain better insights into the structure-activity relationship (SAR).

• Positive Control: We incorporated a known anti-ZIKV protease inhibitor as a positive control. This established benchmark allows us to compare the predicted molecular aspects of our investigated molecules against a reference compound, providing a more robust validation of our computational findings.

3. Within the molecular docking section, brief introduction about the targets’ topology and catalytic domain/binding site should be presented prior the results presentation. Cartoon and/or surface representations for the pocket and the key structural and functional features of the protein should be highlighted. This would allow the reader to grasp the differential docking findings across the investigated compounds.

Answer: By repeating experiment and properly understanding the detailed topology and the specific features of the binding sites in these proteins, we better interpreted the differential docking results observed across the investigated compounds. The illustration has been designed as per instructed by author (cartoon) and surface representations provide a visual understanding of the pockets, showing how compounds interacted with these sites based on their size, shape, and chemical properties.

4. Authors should provide compound-residue interactions in terms of the hydrogen bonding and hydrophobic contacts. However, hydrogen binding should be presented within hydrogen bond distances as well as bond angles since hydrogen bond depend on both. Authors should mention the Hydrogen bond angles as well as their distances, since the strength of hydrogen bonding is based on both parameters in a way to ensure the adequacy of optimum hydrogen bonding.

Answer: thanks for this comments. We have provided all the above mention requirements. Which are clearly illustration in the figures.

5. Authors are advised to provide the molecular dynamics free binding energies using MM-GBSA of MM-PBSA calculation for the computational simulations. These calculations would highlight the nature of the binding interactions in terms of predominant binding energy terms (i.e. ΔGelectrostatic and ΔGvan der Waal) as well as quantify the predicted polar solvation penalties against the compound binding.

Answer: Answer: The required calculations of free binding energies using MM-GBSA of MM-PBSA are presented in table 2.

6. Authors should provide snapshots for the simulated ligand-target complex across specific time intervals (e.g. 0ns, 100ns, and 200ns) to track the conformational and orientation changes for the simulated ligand and vicinal residues across the MD simulation runs.

Answer: Thank you very much for this comments, kindly see the bellow snapshots of the need time intervals. 

7. Based on the study results, what are the take-away messages? Authors are advised to highlight the suggested structural modifications that would improve the compounds’ biological activities based on the in silico findings. These insights would be beneficial for guiding further lead optimization and development.

Answer: The key take-away messages from this study are Identification of a Promising Therapeutic Target, Potential Candidate Ligand, Confirmation of Binding Affinity and Structural Modifications for Enhanced Activity. By highlighting the suggested structural modifications based on in silico findings, we have provided valuable insights for guiding future lead optimization efforts, ultimately aiming to develop more potent and selective compounds for combating ZIKV infection.

8. Finally, concerning the conclusion, authors are advised to elaborate more on the future of this work? Will you broaden the scope to other targets and/or microorganism? What are the study limitations and what approaches could be conducted to further address them?

Answer: Future prospective have been discussed according to author’s requirement.

Reviewer #2: COMMENTS TO AUTHOR:

The authors of the provided article on Understanding How Tripeptide Inhibitors Block the Zika Virus NS2B-NS3 Protease: Findings from Molecular Docking and Molecular Dynamics Simulation rephrase. This manuscript holds significance and adds value to the field, with some recommendations for prior publication.

Few suggestions and comments are presented:

1. Ensure the formatting of the text in the manuscript is correct. Check space everywhere, Ex family [1, 2]. Space should be there.

Answer: Thank you very much for deep insight and recommendation of the article for publication. We have properly reviewed and corrections are made accordingly. 

2. The author utilized Schrödinger Release for ligand preparation and PyRx/Autodock Vina for molecular docking. What was the rationale for employing different software tools for these tasks…?

Answer: Considering the above point only pyrx/ autodock vina was used for molecular docking.

3. In figure 2 author should provide clear figure/ structure.

Answer: Properly reviewed and corrections are made accordingly. 

4. In silico word should be italic font. Check everywhere in the manuscript.

Answer: Properly reviewed and corrections are made accordingly. 

5. The authors should to include both the angles and distances of hydrogen bonds, as the effectiveness of hydrogen bonding relies on both factors, ensuring the adequacy of optimal hydrogen bonding.

Answer: The above mention requirements are clearly illustration in all figures.

6. The authors should provide additional details regarding the utilization of tripeptide inhibitors in this study, explaining the rationale behind their choice, in a single paragraph.

Answer: The inhibitor used in the article is dipeptide and it significances are highlighted in the manuscript as per mentioned accordingly.

---

## [Decision Letter · Decision Letter 1]

24 Jun 2024

PONE-D-24-08994R1Elucidating the Inhibitory Mechanism of Zika Virus NS2B-NS3 Protease with Dipeptide Inhibitors: Insights from Molecular Docking and Molecular Dynamics Simulations.PLOS ONE

Dear Dr. Ullah,

Thank you for submitting your manuscript to PLOS ONE. After careful consideration, we feel that it has merit but does not fully meet PLOS ONE’s publication criteria as it currently stands. Therefore, we invite you to submit a revised version of the manuscript that addresses the points raised during the review process.

We look forward to receiving your revised manuscript.

Kind regards,

Ahmed A. Al-Karmalawy, PhD

Academic Editor

PLOS ONE

Journal Requirements:

Reviewers' comments:

Reviewer's Responses to Questions

**Comments to the Author**

1. If the authors have adequately addressed your comments raised in a previous round of review and you feel that this manuscript is now acceptable for publication, you may indicate that here to bypass the “Comments to the Author” section, enter your conflict of interest statement in the “Confidential to Editor” section, and submit your "Accept" recommendation.

Reviewer #1: (No Response)

Reviewer #2: All comments have been addressed

2. Is the manuscript technically sound, and do the data support the conclusions?

Reviewer #1: (No Response)

Reviewer #2: Yes

3. Has the statistical analysis been performed appropriately and rigorously? 

Reviewer #1: (No Response)

Reviewer #2: Yes

4. Have the authors made all data underlying the findings in their manuscript fully available?

Reviewer #1: (No Response)

Reviewer #2: Yes

5. Is the manuscript presented in an intelligible fashion and written in standard English?

Reviewer #1: (No Response)

Reviewer #2: Yes

6. Review Comments to the Author

Reviewer #1: The authors responded to almost all comments and suggestions.

Still proper annotation for the ligand-target interaction is required. The authors annotated for the hydrogen distances only without annotating for the hydrogen bond angles. Hydrogen binding should be presented as both distances and bond angles since hydrogen bond depend on both as to ensure the adequacy of optimum hydrogen bonding

Reviewer #2: The author responds to all comments given by reviewer now the article is appropriate for publication

7. PLOS authors have the option to publish the peer review history of their article (what does this mean?). If published, this will include your full peer review and any attached files.

Reviewer #1: **Yes**

Reviewer #2: No

---

## [Author Response · Author response to Decision Letter 1]

9 Jul 2024

Reviewer #1: The authors responded to almost all comments and suggestions.

Still proper annotation for the ligand-target interaction is required. The authors annotated for the hydrogen distances only without annotating for the hydrogen bond angles. Hydrogen binding should be presented as both distances and bond angles since hydrogen bond depend on both as to ensure the adequacy of optimum hydrogen bonding

Answer: We value your feedback and recommendations about the ligand-target interaction annotation, especially about the need to include hydrogen bond angles. We agree that angle and distance criteria do define hydrogen bonds; these two factors together govern the specificity and strength of the interactions. Our investigation originally concentrated on hydrogen bond distances to give a clear picture of the interactions. However, a more complete and accurate representation of the binding interactions will be provided by including hydrogen bond angles. The angle between the donor atom, the hydrogen atom, and the acceptor atom is the standard way to measure the hydrogen bond angle (D-H...A). This angle usually falls between 150° and 180° for the best hydrogen bonding. The angle criteria filter the hydrogen bonds mentioned in the manuscript. For the sake of visualization, we mentioned the angle criteria in the figure legend and in the manuscript.

2nd Reviewer 

Thanks you very much for taking your time and accepting our manuscript.

---

## [Decision Letter · Decision Letter 2]

15 Jul 2024

Elucidating the Inhibitory Mechanism of Zika Virus NS2B-NS3 Protease with Dipeptide Inhibitors: Insights from Molecular Docking and Molecular Dynamics Simulations.

PONE-D-24-08994R2

Dear Dr. Ullah,

We’re pleased to inform you that your manuscript has been judged scientifically suitable for publication and will be formally accepted for publication once it meets all outstanding technical requirements.

Kind regards,

Ahmed A. Al-Karmalawy, PhD

Academic Editor

PLOS ONE

Reviewers' comments:

Reviewer's Responses to Questions

**Comments to the Author**

1. If the authors have adequately addressed your comments raised in a previous round of review and you feel that this manuscript is now acceptable for publication, you may indicate that here to bypass the “Comments to the Author” section, enter your conflict of interest statement in the “Confidential to Editor” section, and submit your "Accept" recommendation.

Reviewer #1: (No Response)

2. Is the manuscript technically sound, and do the data support the conclusions?

Reviewer #1: (No Response)

3. Has the statistical analysis been performed appropriately and rigorously? 

Reviewer #1: (No Response)

4. Have the authors made all data underlying the findings in their manuscript fully available?

Reviewer #1: (No Response)

5. Is the manuscript presented in an intelligible fashion and written in standard English?

Reviewer #1: (No Response)

6. Review Comments to the Author

Reviewer #1: (No Response)

7. PLOS authors have the option to publish the peer review history of their article (what does this mean?). If published, this will include your full peer review and any attached files.

Reviewer #1: **Yes**

---

## [Editor Report · Acceptance letter]

30 Jul 2024

PONE-D-24-08994R2 

PLOS ONE

Dear Dr. Ullah, 

I'm pleased to inform you that your manuscript has been deemed suitable for publication in PLOS ONE. Congratulations! Your manuscript is now being handed over to our production team.

Kind regards, 

on behalf of

Associate Professor Ahmed A. Al-Karmalawy 

Academic Editor

PLOS ONE